# Guided Domain Solver: Structured Exploration of Domain-Specific Tasks with Large Language Models

## Abstract

This work presents a method to solve domain-specific problems by leveraging Monte Carlo Tree Search (MCTS), Knowledge Graphs and Large Language Model (LLM) agents. At the core of this approach lies a MCTS algorithm, which explores the complex solution space of a given domain in a goal-directed and sample-efficient manner. In the expansion phase of the MCTS, a domain-specific knowledge graph is incorporated to encode concepts, relationships and constraints. This structured representation enables an LLM agent to make informed decisions for the node expansion. By combining a structured search of the solution space through MCTS, a representation of domain knowledge through the knowledge graph and the generalization abilities of an LLM agent, this method can solve complex tasks in domains where both creativity and adherence to expert rules are essential. In a first step, this approach is used to solve Sokoban, a puzzle game that requires planning and creativity to place several boxes at specific targets with as few moves as possible.

## 1 Introduction

LLMs based on the Transformer architecture (Vaswani et al., 2017) achieve good results in generalizing many tasks and excel particularly in tasks such as text generation or translation. Despite these impressive advances, they tend to hallucinate or have errors in reasoning, especially when performing complex tasks (Chang et al., 2023; Hadi et al., 2023; Ji et al., 2023; Momennejad et al., 2023). In order to improve these shortcomings, several attempts were made to improve the planning of LLMs (Minaee et al., 2025).

Fine-tuning LLMs can be tricky and requires a lot of high-quality data, as well as the construction of a training pipeline, both of which can be costly (Cao et al., 2024). In order to provide the LLM with the latest data without having to retrain it, various connections to databases (Lewis et al., 2021) and search engines (Xiong et al., 2024) were explored.

Deep Reinforcement Learning (DRL) algorithms can be applied for complex planning tasks in an environment. Since rewards are usually sparse in these domains, training is often inefficient. To overcome this, the task is usually divided into sub-goals (Pastukhov, 2025). When changes are made in the environment, DRL agents can lose performance, resulting in retraining or the need for specialized methods (Liu et al., 2023; Chen et al., 2025).

In this work, a method was developed that explores a domain step by step using an MCTS (Coulom, 2007). This allows even complex planning tasks to be addressed. These include tasks such as robotics planning, engineering design automation, puzzle games and scientific discovery. The LLM does not need to be retrained, as recent and enriched data can be embedded in the Knowledge Graph.

Fine-tuning the LLM is expected to result in a more efficient exploration of the solution space. During inference, LLM test-time compute is performed through reasoning and expanding the Knowledge Graph, which can be more effective than scaling model parameters (Snell et al., 2024).

## 2 RELATED WORK

Sokoban is a popular testing ground for approaches that aim to improve the planning capabilities of LLMs. There have been several impressive attempts to solve the game, using LLMs that are retrained on search strategies, as in Searchformer (Lehnert et al., 2024). In this approach, an LLM is fine-tuned on the search dynamic of A* (Hart et al., 1968) to solve Sokoban. Other attempts involve building a world model to solve the game, as in WorldCoder (Tang et al., 2024). Here, knowledge about the game is translated into Python code, which represents the world model. This knowledge is highly interconnected, as in the Knowledge Graph, but is more difficult to expand.

Knowledge Graph combined with LLMs to enhance their capabilities is a promising field of research (Pan et al., 2024; Kau et al., 2024). The field is usually divided into these three categories: Knowledge Graph enhanced LLMs (Lewis et al., 2021; Baek et al., 2023), LLM augmented Knowledge Graph (Hu et al., 2023; Hao et al., 2023), or hybrid approaches (Zhang et al., 2019; Wang et al., 2023).

MCTS has often been used to plan the next action in a large problem space. Examples of this include AlphaZero from Silver et al. (2017) or the External and Internal Planning with Language Models from Schultz et al. (2025).

## 3 GUIDED DOMAIN SOLVER

The Guided Domain Solver uses an MCTS to search a large problem space step by step. Expert knowledge is embedded in a Knowledge Graph and can be used to make the search for promising states more effective. The Knowledge Graph, which is stored in a graph-based database such as Neo4J, contains information about all visited states and the currently selected state. This information is summarized using several Cypher queries to prompt an LLM which evaluates the next action. An overview of the method can be seen in Figure 1.

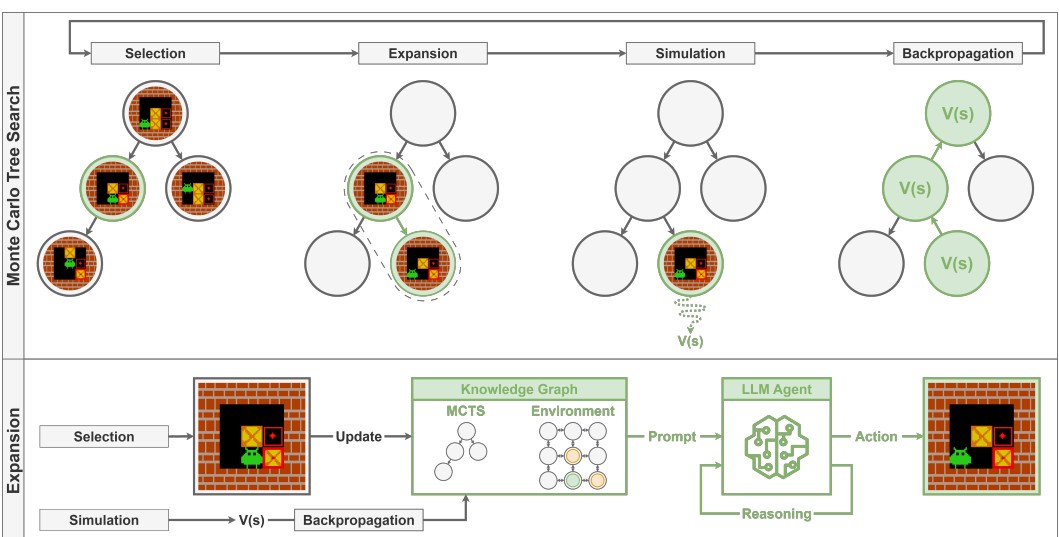

Figure 1: Overview that describes the Guided Domain Solver. The upper half shows the MCTS procedure. The lower half provides a more detailed view of the expansion step, showing how the LLM agent is prompted with the structured information from the Knowledge Graph.

## 3.1 DOMAIN: SOKOBAN

Sokoban is a Japanese puzzle game in which the player must move boxes in a warehouse so that they reach specified target fields. The implementation as OpenAI Gym was created by Schrader (2018). An example of the starting point of the game can be seen on the left side of Figure 2. The player can only move one box at a time. If two boxes are lined up behind each other, they cannot be moved. Dragging boxes is not allowed. A strategic approach is required to avoid maneuvering the boxes into dead ends or blocking the access to the remaining boxes (Racanière et al., 2017).

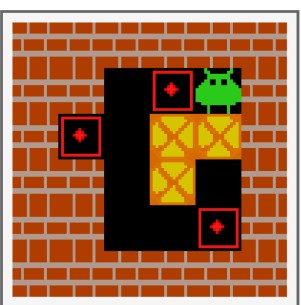 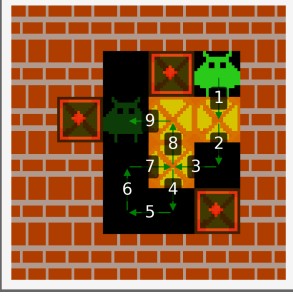

Figure 2: On the left is the starting position of a Sokoban game. To the right is the optimal sequence of actions to solve the game.

The goal is to solve the game in as few steps as possible. An optimal solution looks like the example on the right side of Figure 2. In order to achieve an optimal solution in larger scenes, it is necessary to develop creative long-term strategies.

## 3.2 MONTE CARLO TREE SEARCH

The algorithm is based on MCTS to find a solution for the Sokoban game. It involves the steps of selection, expansion, simulation, and backpropagation, which are repeated several times. The resulting search tree is stored in the Knowledge Graph so that already explored game states can be factored into the expansion of new ones.

### 3.2.1 SELECTION

During the selection phase of MCTS, the algorithm traverses the current search tree to identify the most promising node for further investigation. Normally, an upper confidence bound (UCB) for trees (Auer et al., 2002) is used to find a balance between exploitation and exploration:

$$\text{UCB}_i = V_i + C \cdot \sqrt{\frac{\ln N_i}{n_i}} \tag{1}$$

The exploitation is defined by the value of the state $V_i$. Exploration is controlled by the exploration constant $C$. $N_i$ is the number of visits to the parent node, whereas $n_i$ is the number of visits to the child node. In the case of the Sokoban game, only exploitation is used. This is because it is a fully observable domain and therefore an error-free simulation can be used. Thus, the most promising node that still contains unexplored actions is selected. Once the node has been selected, the Knowledge Graph is updated with the current game status.

### 3.2.2 EXPANSION

In the expansion step, the search tree is extended with a new node. Multiple Cypher queries are made on the Knowledge Graph, which summarize the current state of the game and game states that have already been reached. In addition, the shortest paths to place each of the remaining boxes are calculated. For this purpose, all other boxes are considered walls, which makes some box placements impossible. Missing box placements should be interpolated by the LLM agent using the additional

information. With all this information, an LLM agent is prompted to execute the next action in the game state. An example prompt can be seen in Figure 3.

```
Example Prompt

system:    You are a player who tries to solve a Sokoban game.
           Keep the reasoning short.
           Respond only with a single action out of ['UP', 'DOWN', 'LEFT', 'RIGHT'].

human:     Use the following results retrieved from a database to provide the next
           action for the Sokoban game.
           Environment: {environment}
           Shortest paths to place remaining boxes: {paths_to_place_remaining_boxes}
           Attempted Actions: {attempted_actions}
           Possible Actions: {possible_actions}
           Action:

response:  {action}
```

Figure 3: Example prompt for evaluating the next action in a Sokoban game.

During expansion, it is possible to introduce hard rules using domain knowledge. In the Sokoban game, the actions are mapped to the discrete action space. If the LLM agent does not provide a feasible action, a random possible action is selected as a fallback.

### 3.2.3 SIMULATION

In the simulation phase, the result of the newly added node is evaluated by simulating a trajectory from this point to a final state using an efficient policy. Normally, a ratio of games won to lost is used to determine the value of the game state. Since the Sokoban game has a very sparse distribution of games won, finding a solution using this method can take considerably longer. Here, it is possible to take advantage of the fact that the possible game states of Sokoban are finite. Therefore, the remaining steps to solve the game state are used as the value of the game state. This evaluation is determined using a breadth-first search, which represents an error-free evaluation of the game state that cannot be achieved in many other domains. A description of the breadth-first search algorithm can be found in the Appendix A.

### 3.2.4 BACKPROPAGATION

In the backpropagation step, the result of the simulation is propagated back through the search tree, updating the value estimates of each node along the path. In the case of the Sokoban game, the following update rule is applied here:

$$V(s_t) \leftarrow V(s_t) + \alpha \gamma^n V(s_{t+n}) \tag{2}$$

The update rule is an adapted variation of the $n$-step temporal difference learning approach from Sutton & Barto (2020) without rewards. It gets applied upwards on each node in the search tree to determine the new value of the game state $V(s_t)$. During the update, the number of steps $n$ in the update rule depends on how deep the simulated node is in the search tree. The learning rate $\alpha$ and the discount factor $\gamma$ can be tuned.

### 3.3 KNOWLEDGE GRAPH

The Knowledge Graph allows domain-specific knowledge to be encoded in a structured manner, allowing for precise queries, cross-system integration, and human-interpretable data. This is used to provide the LLM agent with domain knowledge without having to retrain the model. In the case of the Sokoban game, a representation of the game status and the MCTS is stored. The state of the game is divided into *static*, *dynamic*, and *action* layers, as shown in Figure 4. Dividing the Knowledge Graph into layers can be helpful in separating logical structures from one another,

resulting in optimized queries, better maintainability, and improved reasoning. The nodes of the *static* layer are set at the beginning and do not change during the game. All moving objects are mapped in the *dynamic* layer. The nodes in this layer remain throughout the game and only their relations change. The *action* layer determines the possible actions that can be performed in the current state. In any given state, nodes in this layer can disappear or be added.

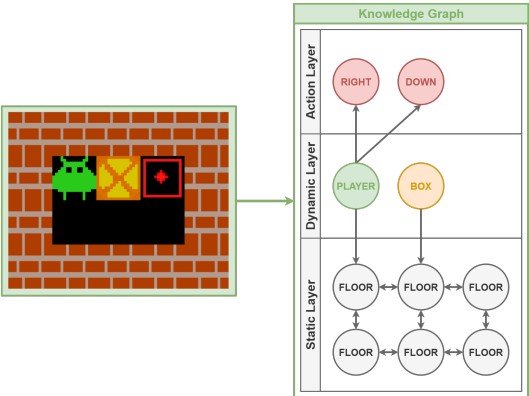

Figure 4: Conversion of the Sokoban game state to a graph representation in the Knowledge Graph.

The nodes of the floors contain the target positions of the boxes as properties. In the relations, the positions of the player and the boxes on the playing field are encoded. This allows effective queries to be performed to determine which box is closest to the player or which box can be placed the fastest. The complete schemas of the Knowledge Graph can be found in the Appendix B.

## 4 EXPERIMENTS

The models Qwen3:32b (Yang et al., 2025), Deepseek-R1:70b (DeepSeek-AI et al., 2025), and GPT-OSS:120b (OpenAI, 2025) were used as LLM agents. These were run locally on an Nvidia RTX A6000. The solutions found for the Sokoban game are equivalent to the optimal solutions, as the same number of steps are required to solve the game. This is made possible by the optimal function within the simulation step. A subset of the solved games can be found in the Appendix C.

Since the optimal solution is always achieved in this domain, the number of nodes required to find the optimal solution is compared here. Therefore, three different methods are evaluated to determine the expansion efficiency as seen in Figure 5. One was expansion with a random possible action. In comparison, random sampling takes a random action from the shortest paths to place the unplaced boxes. The last approach was to use different LLM agents which receives all the information about the environment as well as the shortest paths to place the unplaced boxes.

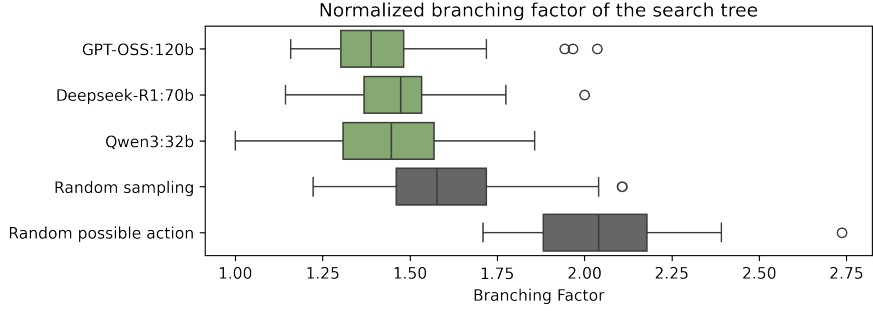

Figure 5: Normalized branching factor across methods, showing that the LLM agent explores fewer nodes relative to solution length, suggesting improved search efficiency.

The plot shows a comparison of the different methods in a greedy selection scenario. An error-free expansion would generate a branching factor of 1, which means that the best action is always taken. It can be seen that the LLM agents are able to make better decisions than the random methods. However, it requires significantly more time for evaluation and reasoning.

## 5 CONCLUSION

The Guided Domain Solver can solve the Sokoban game in an optimal way. Thereby the problem space is searched by the MCTS in an efficient, targeted manner. This initial application demonstrates the powerful combination of MCTS, LLM agent and Knowledge Graph. It paves the way for further applications in other domains that require a balance between flexibility and compliance with constraints. In future work, the method could also be extended to allow the LLM agent to execute its own queries in the Knowledge Graph. In addition, the method could be extended to continuous action spaces or used for continuous domain states.

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

## A  BREADTH-FIRST SEARCH

Breadth-first search is a graph traversal algorithm that explores nodes in order of their depth. It visits all immediate neighbors before moving on to nodes at the next level  (Cormen et al., 2009).

---
**Algorithm 1** Breadth-First Search

---
**Require:** Graph $G = (V, E)$, start node $s$
**Ensure:** Visits all nodes reachable from $s$ in BFS order
 1: Initialize an empty queue $Q$
 2: Mark all nodes as unvisited
 3: Mark $s$ as visited and enqueue it into $Q$
 4: **while** $Q$ is not empty **do**
 5:   $u \leftarrow$ dequeue$(Q)$
 6:   **process** node $u$
 7:   **for all** neighbors $v$ of $u$ **do**
 8:     **if** $v$ is not visited **then**
 9:       Mark $v$ as visited
10:       Enqueue $v$ into $Q$
11:     **end if**
12:   **end for**
13: **end while**

---

## B  KNOWLEDGE GRAPH SCHEMAS

The nodes and their relationships are shown in the schemas of the Knowledge Graph. Following are the schemas of the Knowledge Graph for Sokoban.

### B.1  ENVIRONMENT NODES

```
{
  "Player": {
    "labels": [],
    "properties": {
      "id": {"unique": false, "indexed": false, "type": "INTEGER", "existence": false}
        ,
      "caption": {"unique": false, "indexed": false, "type": "STRING", "existence":
          false},
      "y": {"unique": false, "indexed": false, "type": "INTEGER", "existence": false},
      "x": {"unique": false, "indexed": false, "type": "INTEGER", "existence": false}
    },
    "relationships": {
      "ON_TOP_OF": {
        "direction": "out",
        "labels": ["Floor"],
```

```
486        "properties": {}
487      },
488      "CAN_MOVE": {
489        "direction": "out",
490        "labels": ["Action"],
491        "properties": {}
492      }
493    }
494  },
495  "Box": {
496    "labels": [],
497    "properties": {
498      "id": {"unique": false, "indexed": false, "type": "INTEGER", "existence": false}
499        ,
500      "caption": {"unique": false, "indexed": false, "type": "STRING", "existence":
501          false},
502      "y": {"unique": false, "indexed": false, "type": "INTEGER", "existence": false},
503      "is_on_target": {"unique": false, "indexed": false, "type": "BOOLEAN", "
504          existence": false},
505      "x": {"unique": false, "indexed": false, "type": "INTEGER", "existence": false}
506    },
507    "relationships": {
508      "ON_TOP_OF": {
509        "direction": "out",
510        "labels": ["Floor"],
511        "properties": {}
512      },
513      "SHOULD_GO_TO": {
514        "direction": "out",
515        "labels": ["Floor"],
516        "properties": {}
517      }
518    }
519  },
520  "Floor": {
521    "labels": [],
522    "properties": {
523      "id": {"unique": false,"indexed": false,"type": "INTEGER","existence": false},
524      "has_box_target": {"unique": false,"indexed": false,"type": "BOOLEAN","existence
525          ": false},
526      "caption": {"unique": false,"indexed": false,"type": "STRING","existence": false
527          },
528      "y": {"unique": false,"indexed": false,"type": "INTEGER","existence": false},
529      "x": {"unique": false,"indexed": false,"type": "INTEGER","existence": false}
530    },
531    "relationships": {
532      "ON_TOP_OF": {
533        "direction": "in",
534        "labels": ["Player", "Box"],
535        "properties": {}
536      },
537      "CAN_GO_TO": {
538        "direction": "out",
539        "labels": ["Floor", "Floor"],
         "properties": {}
       },
       "SHOULD_GO_TO": {
         "direction": "in",
         "labels": ["Box"],
         "properties": {}
       }
     }
   }
 }
```

## B.2 MONTE CARLO TREE SEARCH NODES

```
{
  "Path": {
    "labels": [],
    "properties": {
      "id": {"unique": false, "indexed": false, "type": "INTEGER", "existence": false}
        ,
      "possible_actions": {"unique": false, "indexed": false, "type": "LIST", "
        existence": false},
      "reward": {"unique": false, "indexed": false, "type": "FLOAT", "existence":
        false},
      "trajectory": {"unique": false, "indexed": false, "type": "LIST", "existence":
        false},
      "done": {"unique": false, "indexed": false, "type": "BOOLEAN", "existence":
        false},
      "value": {"unique": false, "indexed": false, "type": "FLOAT", "existence": false
        },
      "caption": {"unique": false, "indexed": false, "type": "STRING", "existence":
        false},
      "parent_id": {"unique": false, "indexed": false, "type": "INTEGER", "existence":
        false}
    },
    "relationships": {
      "MOVE": {
        "direction": "out",
        "labels": ["Path"],
        "properties": {
          "id": {"indexed": false, "type": "INTEGER", "existence": false, "array":
            false},
          "caption": {"indexed": false, "type": "STRING", "existence": false, "array":
            false}
        }
      }
    }
  }
}
```

## C SOLVED SOKOBAN GAMES

In the following Figure 6 are several examples of Sokoban games that were solved using the Guided Domain Solver. In each case, the game was solved with the minimum number of steps.

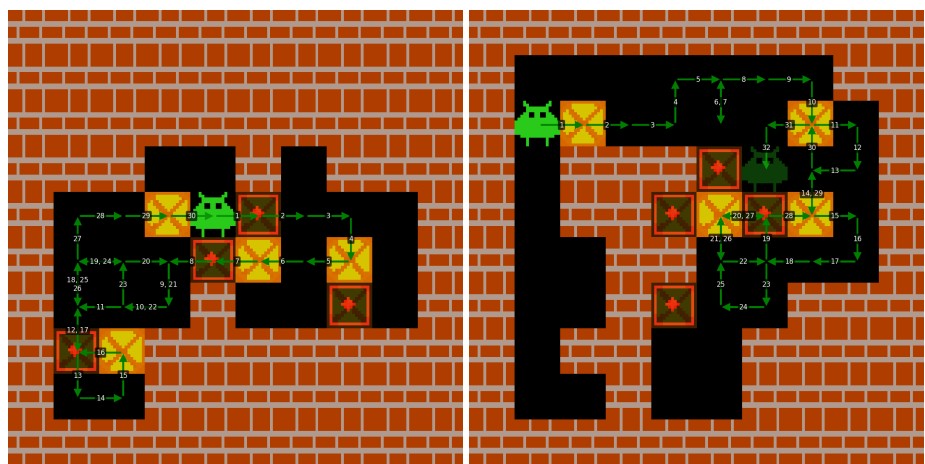

(a) Solved example from Bonnet et al. (2024).    (b) Solved generated Sokoban game.

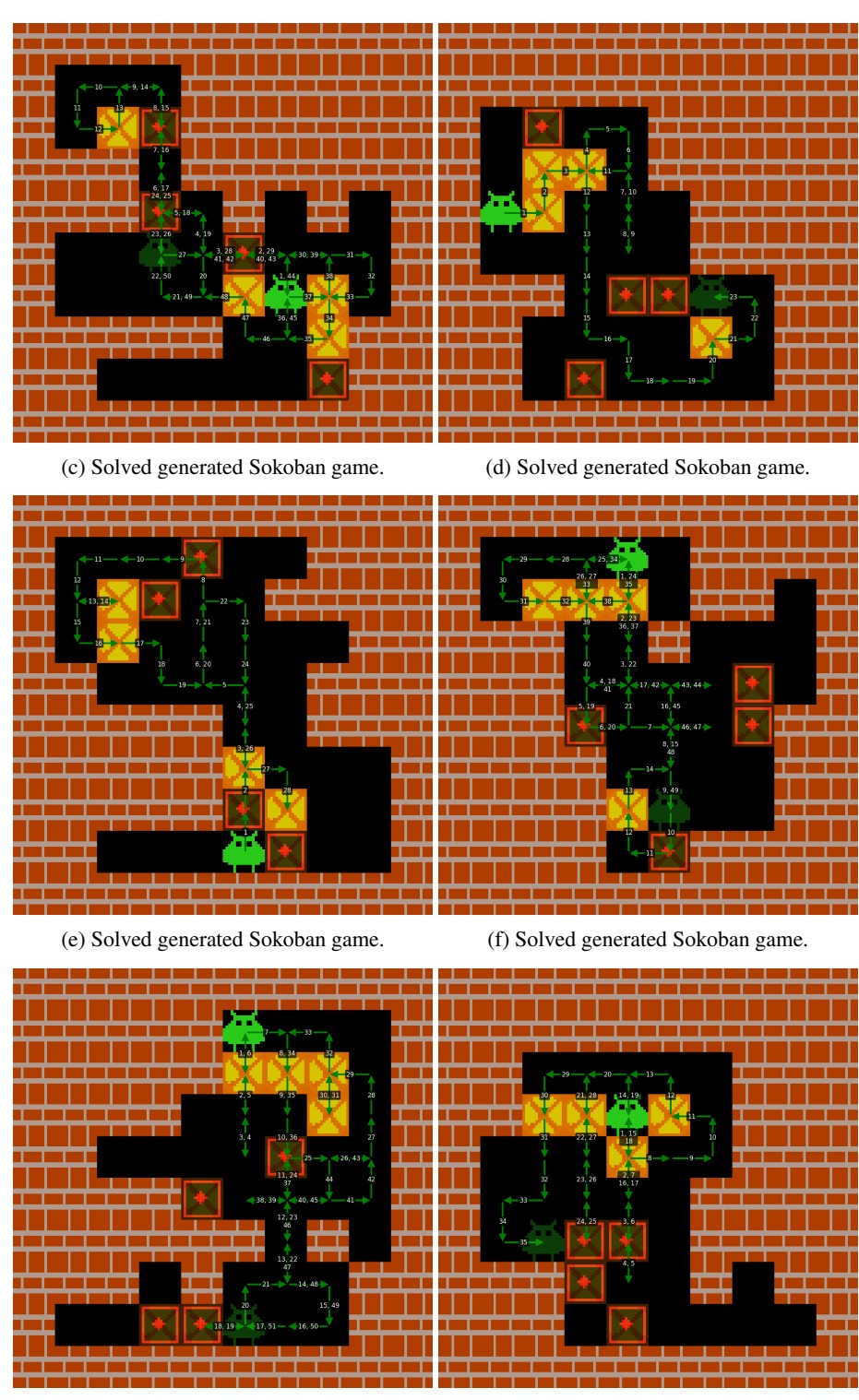

(c) Solved generated Sokoban game.

(d) Solved generated Sokoban game.

(e) Solved generated Sokoban game.

(f) Solved generated Sokoban game.

(g) Solved generated Sokoban game.

(h) Solved generated Sokoban game.

Figure 6: A subset of optimally solved Sokoban games. The images show the start of the game. Numbered green arrows indicate the actions that lead to the solution of the game. The final state of the game is shown in the form of a translucent overlay.

## D    USE OF LARGE LANGUAGE MODELS

During the writing of this paper, LLMs were used as a general-purpose assist tool. Enhancement suggestions from the Overleaf assistant were used to polish the writing style. In addition, DeepL Translator was used for translations or for finding synonyms.

