# OpenReview forum: "Guided Domain Solver: Structured Exploration of Domain-Specific Tasks with Large Language Models"
_ICLR.cc/2026/Conference — ICLR 2026 Conference Withdrawn Submission_

### Official Review · Reviewer_cwyx · 2025-10-24

**Soundness:** 2
**Presentation:** 2
**Contribution:** 2
**Rating:** 4
**Confidence:** 3

**Summary:**

The paper introduces a novel approach to AI planning that combines Monte Carlo Tree Search (MCTS), Knowledge Graphs (KG), and Large Language Models (LLM). The Monte Carlo Tree Search explores the solution space, which is expanded via the concepts/relationships/constraints from a domain-specific KG.

**Strengths:**

The paper's main idea appears to be original. The presentation can be significantly improved, and so does the empirical validation. It is difficult to judge the paper's significance and impact given the relatively small, easy problem instances that it is solving.

**Weaknesses:**

The paper has two main weaknesses: presentation and empirical validation.

With respect to the presentation, the four pages used in current version of Section 3 should be replaced with an illustrative running example (2-3 pages with MCST, KG, Cypher queries, LLM answers) followed by a summary of the current version of the section. All other details could be part of the appendices. For example, Fig 1 is too generic/high-level to help the reader; Figs 3 & 4 can be fully appreciated only if additional figures are added for MCST and KG.

In terms of the empirical evaluation, it is unclear how difficult the problem instances are. The examples shown in the paper seem to have "a floor plan" of at most 50 cells, which is extremely small. Can GDS tackle floor plans of - say- 1K or 10K or 100K cells? How about with various "obstacles to be removed to create a passage" and "tricky straights to be navigated?" Ideally, the empirical validation should be expanded to a more comprehensive section with increasingly larger "floor spaces" and harder navigation problems. For example, see the [Muslea, 1997] paper below, which is also evaluated on a Shokoban-like world.

Muslea, Ion. "SINERGY: A linear planner based on genetic programming." In European Conference on Planning, pp. 312-324. Berlin, Heidelberg: Springer Berlin Heidelberg, 1997.

**Questions:**

1. line 250 - what do you mean when you claim "the optimal solution is always achieved in this domain"? What guarantees the optimality of GDS's solutions? Per lines 181-183, optimality does NOT seem to be guaranteed.
2. What happens if a problem instance has no solution? Will GDS search "forever," or does it have any way detect such situations?

---

### Official Review · Reviewer_JiUF · 2025-10-30

**Soundness:** 2
**Presentation:** 3
**Contribution:** 1
**Rating:** 0
**Confidence:** 5

**Summary:**

The paper proposes an approach to solve the Sokoban game by integrating LLMs, MCTS, and a Knowledge Graph that basically maintains the current state of the game. The game is basically expanded using MCTS, where the LLM is used to select which action to perform from the child nodes of the nodes selected for expansion by the MCTS. Experiments show improved performance compared to random sampling.

**Strengths:**

Sokoban is a nice problem to work on and integrating MCTS and LLMs to solve puzzles is interesting.

**Weaknesses:**

The motivation to the paper is not clear to me. If we know the rules and dynamics of the game, why use an LLM? One can simply use classical planning to solve this problem, i.e., encode it in PDDL and run a planner. This has been done for Sokoban for many years.
I recommend the authors to look deeper to the automated planning community’s work on this puzzle and explain how it relates to your work. They seem to solve the same problem.
Only solving Sokoban is, I believe, too limited for a top conference paper.
The baseline of random sampling is too weak.

**Questions:**

1.	What is a “Cypher query”? (line 84)
2.	What is the advantage of using your approach compared to classical planning for this domain?
3.	When you say that you always return optimal solutions, what is the optimality criteria you refer to? Do you mean minimal number of steps? If so, I am not convinced you can really guarantee this. Can you?
4.	Fig. 5 shows normalized branching factor – this is a weird choice. Why not show either number of nodes expanded or solution length? Which is more common for this problem.

---

### Official Review · Reviewer_Vvm6 · 2025-10-31

**Soundness:** 1
**Presentation:** 2
**Contribution:** 1
**Rating:** 2
**Confidence:** 4

**Summary:**

This paper presents a method that combines Monte Carlo Tree Search (MCTS), Knowledge Graphs, and Large Language Models
 (LLMs). The approach uses MCTS to explore the solution space, while a domain-specific Knowledge Graph encodes domain knowledge.

The method is demonstrated on the Sokoban puzzle game, where it achieves optimal solutions by combining error-free simulation with LLM-guided expansion. Experiments show that LLM agents, when guided by the Knowledge Graph, explore fewer nodes than random baselines, indicating improved search efficiency.

It can be splitted into 4 steps: Selection (most promissing node), Expansion (extend to new node),
Simulation (new node evaluation), Backpropagation (update nodes).

**Strengths:**

Incorporation of structured domain knowledge without retraining LLMs.

**Weaknesses:**

- The text needs a few improvements to justify proposed methodology, novelty, approach, contributions.  First few paragraphs were quite vague on what exactly this method is trying to solve and what is the main contributions.
- Experiments shows improved efficiency, but the term is not formally defined. The experiments are very limited.
- Abstract mentions 'creativity', but these are not referred again on the text, or quantified in any way.
- Strong conclusions from limited experimentation - conclusion MCTS is efficient and targeted  based only on one set of experiments on a single domain (Sokoban).
- Lack of baselines - there are other methods with similar approaches (combine graphs and LLMs),  yet there are no baselines added for comparison besides random.
- Overall paper is very short, and a lot more about methodology and contributions could have been added.

**Questions:**

- Your conclusions about efficiency and targeted search are based solely on Sokoban.  How do you expect the method to perform in less structured, partially observable, or continuous domains?  Have you considered or attempted any experiments outside Sokoban to support your claims of generality?
- Why were no strong baselines included, such as other LLM+graph or LLM+search approaches?  How does your method compare to recent work?
- How scalable is the knowledge graph construction and querying process as domain complexity grows?

---

### Official Review · Reviewer_k6mZ · 2025-11-01

**Soundness:** 1
**Presentation:** 1
**Contribution:** 1
**Rating:** 2
**Confidence:** 4

**Summary:**

The paper solves domain-specific problems by leveraging MCTS, knowledge graphs, and LLM agents. An initial experiment was conducted on Sokoban to show the method helps find the solution more quickly.

**Strengths:**

The direction of solving complex planning tasks has high potential.

**Weaknesses:**

The paper is still at a preliminary stage. Here are some major suggestions to improve the work in the future:

Include other tasks, metrics, and more reasonable baselines to make the experiment solid. The authors can refer to relevant literature for some design choices.

I think the paper can benefit from better motivating the method choice. Like why do you need knowledge graphs or MCTS in the first hand?

**Questions:**

See those in the weakness section.

---

### Official Review · Reviewer_613i · 2025-11-02

**Soundness:** 2
**Presentation:** 2
**Contribution:** 2
**Rating:** 0
**Confidence:** 4

**Summary:**

The paper introduces an innovative framework that integrates MCTS, Knowledge Graphs, and LLMs to solve complex domain-specific problems. The approach leverages MCTS for exploration, a knowledge graph for structured domain representation, and LLM reasoning for informed decision-making during search expansion. Validated on the Sokoban puzzle game, the system achieves improved search efficiency and interpretability without retraining the LLM.

**Strengths:**

I really like the core idea of this paper i.e. combining symbolic reasoning (MCTS and KG) with LLM-based semantic understanding. The framework bridges structured search and flexible reasoning, showing how rule-based exploration can be guided by high-level language reasoning.

**Weaknesses:**

The paper feels rather rushed and incomplete. The related work section is shallow and does not sufficiently position this method within existing literature on LLM-based planning or neuro-symbolic systems. The experimental setting is narrow and limited to a single environment (Sokoban)  and the baseline comparisons are weak. Moreover, the paper lacks ablation studies to disentangle the contribution of each component (MCTS, KG, LLM).

**Questions:**

- The prompt template in Figure 3 is helpful but overly simplistic; it doesn’t show how the LLM integrates graph-based reasoning.
- The results section contains only one plot (Figure 5) comparing normalized branching factors. There are no quantitative metrics

---

### Note · Authors · 2025-11-14

**Comment:**

We appreciate the reviewers' efforts and feedback, which will serve as a guide for our future revisions. Therefore, we are withdrawing the paper.

**Withdrawal Confirmation:**

I have read and agree with the venue's withdrawal policy on behalf of myself and my co-authors.